# Nuclear envelope deformation controls cell cycle progression in response to mechanical force

Julien Aureille[1] , Valentin Buffière-Ribot[1], Ben E Harvey[1], Cyril Boyault[1], Lydia Pernet[1], Tomas Andersen[2], Gregory Bacola[3], Martial Balland[2], Sandrine Fraboulet[1], Laurianne Van Landeghem[3] & Christophe Guilluy[1,*]

## Abstract

The shape of the cell nucleus can vary considerably during developmental and pathological processes; however, the impact of nuclear morphology on cell behavior is not known. Here, we observed that the nuclear envelope flattens as cells transit from G1 to S phase and inhibition of myosin II prevents nuclear flattening and impedes progression to S phase. Strikingly, we show that applying compressive force on the nucleus in the absence of myosin II-mediated tension is sufficient to restore G1 to S transition. Using a combination of tools to manipulate nuclear morphology, we observed that nuclear flattening activates a subset of transcription factors, including TEAD and AP1, leading to transcriptional induction of target genes that promote G1 to S transition. In addition, we found that nuclear flattening mediates TEAD and AP1 activation in response to ROCK-generated contractility or cell spreading. Our results reveal that the nuclear envelope can operate as a mechanical sensor whose deformation controls cell growth in response to tension.

**Keywords** AP1; c-Jun; mechanotransduction; nuclear envelope; TEAD
**Subject Categories** Cell Adhesion, Polarity & Cytoskeleton; Cell Cycle

## Introduction

Whether it is generated by myosin motors or externally applied, cells are constantly subjected to mechanical tension, which profoundly impacts their growth [1–3]. Mechanical tension triggers changes in cell shape, adhesion, and cytoskeletal structures that control the cell cycle machinery and orchestrate cellular rearrangements for division [1,4–6]. As cells experience tension, the nucleus undergoes significant morphological changes due to its connection with the cytoskeleton that transmits mechanical stress to the nuclear envelope [7,8]. The inner nuclear membrane interacts with a meshwork of intermediate filaments, i.e., the lamina, whose mechanical properties determine the nuclear strain in response to mechanical stress emanating from the microenvironment [9] or generated by the cytoskeleton [7,10]. Accordingly, abnormal nuclear shapes can be observed in diseases when the lamina is altered, including cancer [11,12] and laminopathies [13], or when there is an increase in mechanical stress, for example, in the arterial wall during hypertension [14]. Whereas changes in nuclear shape can occur during developmental and pathological processes [11,12,15], little is known regarding the impact of nuclear morphology on cell behavior.

External mechanical forces can trigger nuclear envelope (NE) remodeling, which in turn impacts nuclear structure and function [15,16]. Recent studies indicate that this response is partly mediated by mechanosensitive mechanisms located at the NE [17–19]. For example, it has been reported that tension within the NE can regulate gene expression through nuclear pore complex (NPC)-dependent YAP nuclear entry [20] or phospholipase A2 activation [17]. However, when cells are subjected to mechanical stress, many force-bearing elements participate in the cellular response [21,22]. The mechanosensitive processes mediated by the NE can be difficult to distinguish from those controlled by cytoplasmic myosin II-dependent pathways. In this study we combined biophysical and molecular biology approaches, to manipulate the nucleus morphology independently of myosin II activity and we investigated the molecular pathways activated by nuclear deformation.

## Results

### Nuclear flattening is necessary for G1 to S transition

We analyzed the nuclear morphology over the course of the cell cycle using HeLa cells expressing the fluorescence ubiquitination-based cell cycle indicator (FUCCI), and we measured cell (Fig 1A) and nuclear height (Fig 1B and C). We found that cells decrease their height during G1 phase and displayed a flatter nucleus before

1   Institute for Advanced Biosciences, Centre de recherche UGA – INSERM U1209 – CNRS UMR 5309, Grenoble, France
2   Laboratoire Interdisciplinaire de Physique, UMR CNRS 5588, Université Grenoble Alpes, Grenoble, France
3   Department of Molecular Biomedical Sciences, College of Veterinary Medicine, North Carolina State University, Raleigh, NC, USA
    *Corresponding author. Tel: +33 4 7654 9573; E-mail: christophe.guilluy@inserm.fr

transitioning from G1 to S, compared to cells in early G1 or G2 phase (Fig 1A–C). We then investigated nuclear morphology in other cell lines after synchronization, and we observed similar significant decrease in nuclear height during G1 in MEFs and MRC5 cells (Fig 1D and E), indicating that nuclear flattening may occur in several adherent cells. To prevent nuclear flattening in proliferating cells, we used blebbistatin a myosin II inhibitor, which induced an increase in nuclear height due to the disassembly of perinuclear actin cap filaments [7] (Fig 1F and G). We found that myosin II inhibition caused a significant decrease in cells progressing into S and G2 phases (Fig 1H and I). We obtained similar results when we quantified incorporation of the modified thymidine analogue EdU to analyze S phase completion by flow cytometry (Fig 1J and K). This is consistent with previous work showing that cell-generated tension promotes G1 to S transition [1,23,24]. We next wanted to determine whether nuclear flattening would be sufficient to restore cell cycle progression in the absence of mechanical tension. To do so, we adapted a previously developed method [17,25] to modulate nuclear height independently of myosin activity. We applied an agarose pad on top of blebbistatin-treated cells to flatten their nuclei and restore a nuclear morphology similar to those of control cells (Figs 1F and G, and EV1A and B). Using this approach, we did not observe any significant changes in nuclear volume (Fig EV1B) or any nuclear herniation (Fig EV1D) and cell survival was unaffected by the mechanical constraint exerted on the nucleus (Fig EV1C). Strikingly, we observed that compression of the nucleus of blebbistatin-treated cells was sufficient to restore G1 to S transition (Fig 1H–K).

## Nuclear shape regulates transcription factor activity

We next wanted to investigate how nuclear flattening can promote cell cycle progression. Since previous work has shown that tension within the NE can regulate gene expression [17,20], we hypothesized that changes in nuclear shape may regulate the expression of genes that contribute to G1/S transition. To test this idea, we modulated nuclear height in the absence of myosin II activity using the agarose pad (Figs 1F and G, and EV1A) and we analyzed the effect of nuclear shape on the transcriptional machinery. Application of myosin-dependent tension on integrin-based adhesion activates mechanotransduction mechanisms, which mediate cytoskeletal rearrangement, adhesion growth [26] and ultimately regulate gene expression [21]. To ensure that manipulating nuclear morphology did not impact adhesion mechanotransduction mechanisms, we analyzed adhesion size and adhesion protein phosphorylation. As expected, blebbistatin treatment decreased adhesion area (Fig EV1E) and we did not observe any significant difference in adhesion size, adhesion protein tyrosine phosphorylation (Fig EV1F), nor paxillin phosphorylation (Fig EV1G) when the nuclei of blebbistatin-treated cells were mechanically constrained. This indicates that adhesion mechanotransduction mechanisms are not activated in response to changes in nuclear shape and thus should not have any impact on transcriptome in our experimental set-up. We then measured the activity of 345 transcription factors (TFs) by probing their ability to bind to their target DNA sequences (Figs 2A and EV3A and B). We found that blebbistatin treatment inhibited 17 TFs (Fig 2A), including TEAD and HNF4 as reported by previous studies [27,28]. We reasoned that TFs whose activity was directly impacted by nuclear shape would have distinct activity

levels in flat nuclei (Ctrl and Blebb + AP) versus non-flattened nuclei (Blebb). We found 15 TFs that were activated in flat nuclei but not in the nuclei of blebbistatin-treated cells (Fig 2A). In order to identify those responsible for significant transcriptomic consequences in response to nuclear constraint, we selected genes known to be positively regulated by these 15 TFs using the TRRUST database [29] and we analyzed their mRNA expression (Figs 2B and EV3C). Interestingly, only the genes targeted by AP1, TEAD, PPAR, or SP1 showed significant increased expression in response to nuclear flattening (Figs 2B and EV3C). Additional transcriptional regulatory mechanisms, such as epigenetic silencing of the target promoter or coactivator requirement, may explain why we did not detect any change in mRNA expression of the genes regulated by the other TFs (Fig EV3C). Our results reveal that agarose pad-induced nuclear flattening is sufficient to activate AP1, TEAD, PPAR, and SP1 in the absence of myosin II-dependent contractility. However, we cannot rule out the possibility that cytoplasmic mechanotransduction pathways may also contribute to activate these TFs, considering the cellular deformation that occurs following agarose pad application. Since the nucleus undergoes significant morphological changes in response to mechanical stress [7,9], we hypothesized that these 4 TFs may interact with proteins related to mechanical stress. To test this hypothesis, we performed a bioinformatics analysis using the Biological General Repository for Interaction Datasets (BIOGRID) database [30], which annotates protein and genetic interactions. We obtained a minimal network connecting AP1, TEAD, PPAR, and SP1 with a group of proteins involved in the cellular response to mechanical stress, such as PRKM3 (Fig EV3D). This suggests that these TFs may be regulated by signaling pathways activated by mechanical stress.

## Nuclear flattening promotes cell cycle progression via TEAD and AP1

A recent report has shown that AP1 and TEAD can act synergistically on promoter regions and activate a set of target genes that controls cell cycle progression [31]. This suggests that nuclear flattening may trigger a transcriptional program that promotes cell cycle progression via the dual activation of TEAD and AP1. We decided to focus on these two TFs. Firstly, we tested the robustness of our results using different approaches to manipulate the nuclear height. We cultured cells on matrix with different rigidities, as cells generate more perinuclear actin filaments which flatten their nuclei on stiff surfaces [32] (Fig EV3E). As expected, we found that TEAD and AP1 are more active in cells cultured on stiff versus soft matrix (Fig 2C and D). The perinuclear actin filaments are connected to the nuclear envelope by the linker of nucleoskeleton and cytoskeleton (LINC) complex, which allows force transmission to the nucleus and regulates its shape [7]. In order to disrupt the LINC complex, we depleted both SUN1 and SUN2 (Figs EV3H and I, and EV4E), which lead to an increase in nuclear height (Fig EV4G). Consistent with our previous results, this increase in nuclear height was associated with a decrease in AP1 and TEAD activity (Fig EV3H and I). We next plated cells on poly-L-Lysine-coated surfaces to impede adhesion and perinuclear actin filament formation, as demonstrated by their increased nuclear height (Fig EV2A and B). We then used an atomic force microscope to apply compressive force on the nucleus and maintain a constant height for 120 min (Fig EV2A). We

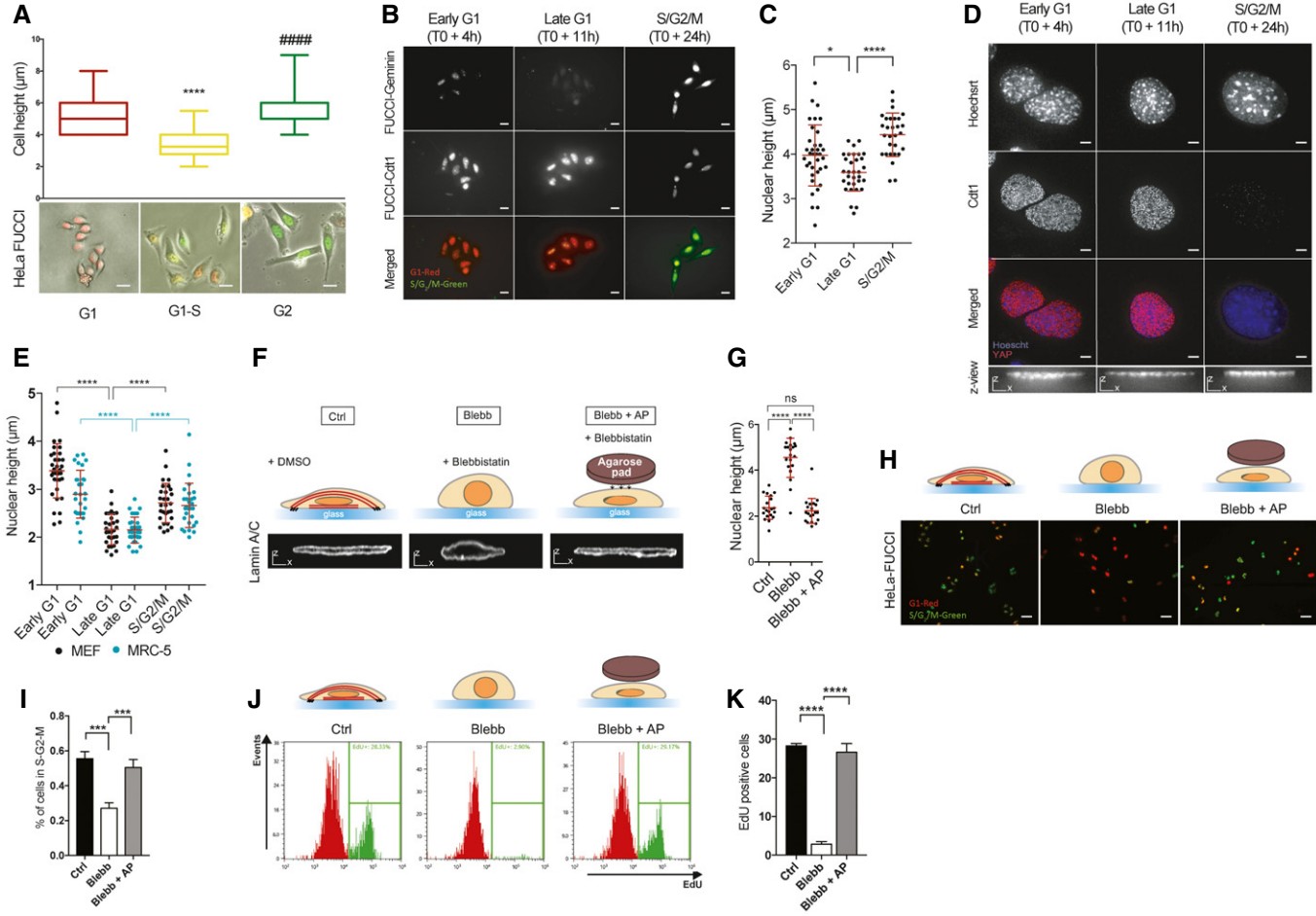

**Figure 1. Nuclear flattening is necessary for G1 to S transition.**

A  Cell height was measured using AFM for HeLa-FUCCI in the indicated phase of mitosis. Below: representative images of synchronized HeLa-FUCCI (box plot, median min/max, n = 20 minimum, ****, ####P < 0.001 one-way ANOVA—Tukey's multiple comparisons post-test, scale bar 15 μm).

B  Representative images of synchronized HeLa stained with an anti-geminin to confirm the indicated phase of mitosis (scale bar 15 μm).

C  Nucleus height was measured by immunofluorescence using Hoechst staining (n = 29 minimum for nucleus height, *P < 0.5, ****P < 0.001 one-way ANOVA—Tukey's multiple comparisons post-test).

D  Representative images of synchronized MEFs stained with an anti-Cdt1 to confirm the indicated phase of mitosis (scale bar 5 μm, bottom scale bar x = 5 μm z = 5 μm).

E  Nucleus height of synchronized MEFs (black dot) and synchronized MRC-5 (blue dot) was measured by immunofluorescence using Hoechst staining. Data are presented as mean ± s.e.m. (n = 29 minimum for nucleus height, ****P < 0.001 one-way ANOVA—Tukey's multiple comparisons post-test).

F  Schematic representation of the method used to induce changes in nuclear shape independently of the actomyosin cytoskeleton. See Fig EV1 for more details. Briefly, cells are treated with DMSO (Control-Ctrl), blebbistatin only (Blebb), and blebbistatin whereas an agarose pad (AP) is used to flatten their nuclei and restore a nuclear morphology similar to those of control cells (Blebb + AP). XZ view of representative nuclei (lamin A/C) is shown in these three conditions (scale bar x = 5 μm z = 5 μm).

G  Nucleus height was measured by immunofluorescence using lamin A/C staining. Data are presented as mean ± s.e.m. (n = 20 for nucleus height, ****P < 0.001 one-way ANOVA—Tukey's multiple comparisons post-test).

H  Representative HeLa-FUCCI cells in S-G2-M (green) and G1 phase (red) in Ctrl, Blebb, and Blebb + AP conditions. Scale bar = 31 μm.

I  Corresponding percentage of HeLa-FUCCI in mitosis. Data are presented as mean ± s.e.m. (n = 24 fields minimum from four independent experiments, ***P < 0.01 one-way ANOVA—Tukey's multiple comparisons post-test).

J  Representative flow cytometry histograms for EdU-positive cells in Ctrl, Blebb, and Blebb + AP conditions.

K  Corresponding percentage of EdU-positive cells. Data are presented as mean ± s.e.m. (n = 4 independent experiments with at least 60,000 events for each condition per experiment. ***P < 0.001 one-way ANOVA—Tukey's multiple comparisons post-test).

observed a significant activation of TEAD and AP1 in response to nuclear compression (Fig EV2C and D), confirming our previous observations. To test whether TEAD and AP1 could impact cell cycle progression in response to nuclear flattening, we analyzed the expression of their target genes involved in proliferation (Fig 2E).

mRNA expression of these target genes mirrored TEAD and AP1 transcriptional activity (Fig 2A) and was increased in response to nuclear compression in blebbistatin-treated cells (Fig 2E). AP1 is composed of the dimers of Fos, Jun, as well as ATF family members [33]. We observed that depletion of c-Jun prevented AP1 activation

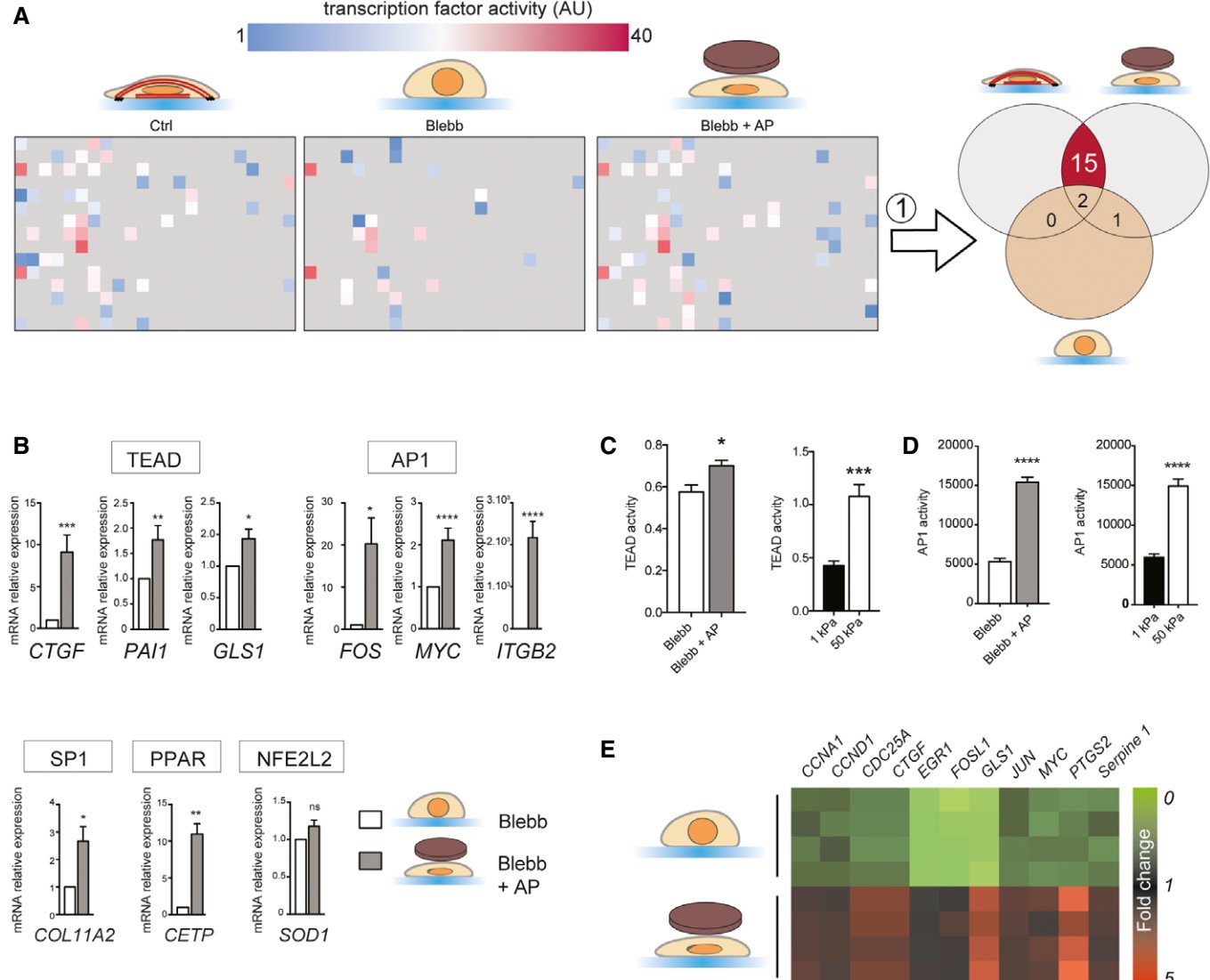

**Figure 2. Nuclear shape regulates transcription factor activity.**

A  Analysis of transcription factor activity (Affymetrix combo protein–DNA array) for Ctrl, Blebb, and Blebb + AP conditions. Each square corresponds to a specific transcription factor (TF) activity and is color coded to show the relative activity. To identify TFs regulated by nuclear shape, we selected TFs that were more active in Ctrl and Blebb + AP compared to blebbistatin-treated cells (Blebb).

B  TF's target genes and GAPDH mRNA levels were analyzed by real-time qPCR. Results are expressed as relative mRNA expression levels. Data are presented as mean ± s.e.m. (n = 4 minimum, t-test *P < 0.5, **P < 0.05, ***P < 0.01, ****P < 0.001).

C  TEAD activity was analyzed in HeLa cells co-transfected with a Renilla plasmid as a luciferase reporter plasmid controlled by the TEAD-responsive promoter, and with a Renilla plasmid as a gene reporter; HeLa cells were cultured on 1 and 50 kPa (n = 3, t-test ***P < 0.01) or treated with blebbistatin and blebbistatin + AP (n = 6; t-test *P < 0.5). Data are presented as mean ± s.e.m.

D  AP1 activity was assessed in the indicated conditions using AP1 reporter assay. Data are presented as mean ± s.e.m. (n = 5, t-test ****P < 0.001).

E  Heatmap of mRNA expression profiles of genes regulated by TEAD and AP1.

in blebbistatin-treated cells whose nuclei were mechanically constrained (Fig EV3F). TEAD is the downstream target of YAP, a transcription regulator that coactivates and associates with TEAD upon translocation into the nucleus [27]. To test the involvement of AP1 and TEAD during cell cycle progression, we depleted YAP and c-Jun. We found that nuclear flattening had no effect on S phase completion in cells depleted in YAP and c-Jun, indicating that AP1 and TEAD are required for nuclear compression-induced G1/S transition (Fig EV3G).

**Nuclear flattening is sufficient to activate c-Jun and YAP independently of actomyosin contractility and cell spreading area**

We next wanted to investigate how nuclear compression activates AP1 and TEAD. Since the transcriptional activity of c-Jun is regulated by phosphorylation at Ser63 and Ser73 [33], we analyzed c-Jun phosphorylation by immunofluorescence. Similarly to AP1, we found that c-Jun nuclear phosphorylation on

Ser63/73 was both decreased in blebbistatin-treated cells and increased in response to nuclear flattening (Figs 3A and B, and EV4A and B), indicating that c-Jun phosphorylation mediates AP1 activation in response to nuclear shape. To measure YAP activity, we analyzed its nucleo-cytoplasmic localization. As shown by others [27], YAP nucleo-cytoplasmic ratio was decreased in response to myosin II inhibition (Fig 3C and D). Remarkably, mechanical nuclear flattening of blebbistatin-treated cells induced an increase in YAP nucleo-cytoplasmic ratio, which reached similar values to those observed in control cells (Figs 3C and D, and EV4C). To analyze the effect of an increase in nuclear height on YAP localization and c-Jun phosphorylation, we depleted SUN1 and SUN2 (Figs 3G and EV4E and F) and observed a decrease in c-Jun nuclear phosphorylation on Ser63 and YAP nuclear–cytoplasmic ratio (Figs 3G and EV4E and F). Since changes in cell shape and mechanical tension are known to activate c-Jun and YAP [27], we next wanted to test whether nuclear flattening was sufficient to activate these pathways independently of actomyosin contractility or cell spreading area in our experimental system. We first used pharmacological drugs affecting ROCK activity (Y27632) and actin polymerization (cytochalasin D). We observed an increase in nuclear height (Fig EV4G) and a decrease in both c-Jun phosphorylation and YAP nucleo-cytoplasmic localization (Figs 3E and F, and EV4D) in cells treated with either ROCK inhibitor or cytochalasin D. This effect was reversed by mechanical compression of the nuclei (Fig 3E and F). To manipulate cell shape, we plated cells on micropatterned surfaces of 500 or 1,500 $\mu m^2$. Cells on small surface displayed a significantly higher nucleus, and we observed a decreased c-Jun phosphorylation and YAP nuclear–cytoplasmic ratio in these cells compared to cells on 1,500 $\mu m^2$ surfaces (Fig 3H and I). Strikingly, nuclear flattening of cells plated on 500 $\mu m^2$ induced by the agarose pad led to a significant increase in YAP and c-Jun activity, which reached levels similar those observed in cells on 1,500 $\mu m^2$ (Fig 3H and I). These results indicate that nuclear compression is sufficient to activate YAP and c-Jun activation independently of actomyosin contractility and cell spreading area.

## TEAD and AP1 activation are distinctively controlled by nuclear shape

We next wanted to explore the molecular mechanisms mediating YAP and c-Jun activation in response to changes in nuclear shape. Previous studies have shown that mechanical tension activates YAP through mechanisms either dependent or not of the Hippo signaling pathway [27,34–36]. More recently, it was reported that force applied on the nucleus increases YAP nuclear translocation by regulating nuclear pore transport, independently of YAP phosphorylation [20]. Pharmacological inhibition of importin-β-mediated nuclear import decreased drastically YAP translocation (Figs 4A and EV5A) and nuclear phosphorylated c-Jun (Fig EV5B). Even though we were still able to detect a significant increase in YAP nuclear–cytoplasmic localization in response to nuclear flattening, one could anticipate that this increase may not be biologically relevant (Fig 4A). In parallel, we analyzed the phosphorylation of Hippo signaling actors, such as YAP, its upstream kinase large tumor suppressor (LATS), mammalian Ste20-like kinases (MST), Kibra, and PTPN14. Whereas we found that YAP and LATS phosphorylation were increased in blebbistatin-treated cells and decreased in response to nuclear flattening (Fig 4B–D), we observed no difference in MST phosphorylation (Fig 4B). To test the involvement of LATS, we expressed wild type or phosphomimetic mutant (T1079E) of LATS (Fig EV5C–E). The expression of WT or T1079E LATS mutant did not affect nuclear height (Fig EV5E), but decreased YAP nucleo-cytoplasmic ratio in control cells (Fig EV5D). However, it did not affect YAP nucleo-cytoplasmic localization when the nuclei of blebbistatin-treated cells were mechanically constrained (Fig EV5D). This suggests that another kinase or phosphatase specific for YAP may regulate its activity in response to nuclear compression. Another possibility is that nuclear flattening may trigger two mechanisms, one involving YAP phosphorylation and another independent of YAP phosphorylation status which could override YAP phosphoregulation allowing its translocation in response to nuclear flattening. Such mechanism has been suggested by others [34] and may involve regulation of nuclear pore complex by mechanical stress as described recently [20].

**Figure 3. Nuclear flattening is sufficient to activate c-Jun and YAP independently of actomyosin contractility and cell spreading area.**

A  Representative cells stained for p-Jun Ser63 (magenta) and for DNA (cyan) in Ctrl, Blebb, and Blebb + AP conditions. Scale bar = 10 μm. Nuclear heights were measured using Hoechst staining.

B  Corresponding quantifications of p-Jun Ser63 nuclear intensity. Data are presented as mean ± s.e.m. ($n$ = 59 minimum from two independent experiments, ****$P$ < 0.001 one-way ANOVA—Tukey's multiple comparisons post-test).

C  Representative cells stained for YAP (magenta) and for DNA (cyan) in Ctrl, Blebb, and Blebb + AP conditions. Scale bar = 10 μm. Nuclear heights were measured using Hoechst staining.

D  Corresponding quantifications of YAP activity (ratio nucleus/cytosol). Data are presented as mean ± s.e.m. ($n$ = 82 minimum from four independent experiments, ****$P$ < 0.001 one-way ANOVA—Tukey's multiple comparisons post-test).

E  Quantifications of YAP activity (nucleo-cytoplasmic ratio) and p-Jun Ser63 nuclear intensity in Ctrl, Blebb, and Blebb + AC condition and in Ctrl, Y27632, and Y27632 + AP. Data are presented as mean ± s.e.m. ($n$ = 19 minimum, ****$P$ < 0.001 Tukey's multiple comparisons post-test).

F  Quantifications of YAP activity (nucleo-cytoplasmic ratio) and p-Jun Ser63 nuclear intensity in Ctrl, cytochalasin D (CytoD), and CytoD + AP condition. Data are presented as mean ± s.e.m. ($n$ = 32 minimum from two independent experiments, ****$P$ < 0.001 Tukey's multiple comparisons post-test).

G  Quantifications of YAP activity (nucleo-cytoplasmic ratio) and p-Jun Ser63 nuclear intensity in cells depleted or not for SUN1 and SUN2. Data are presented as mean ± s.e.m. ($n$ = 28 minimum, $t$-test, ****$P$ < 0.001).

H  Representative cells cultured on circular micropatterns with surfaces of 1,500 and 500 $\mu m^2$ and stained for p-Jun Ser63 (magenta), for YAP (cyan), and for DNA (green). Additionally, cells were cultured on the micropatterns of 500 $\mu m^2$ and an agarose pad was used to flatten their nuclei. Scale bar = 10 μm. Nuclear heights were measured using Hoechst staining.

I  Corresponding quantifications of YAP activity (nucleo-cytoplasmic ratio) and p-Jun Ser63 nuclear intensity. Data are presented as mean ± s.e.m. ($n$ = 15 minimum ****$P$ < 0.001 one-way ANOVA—Tukey's multiple comparisons post-test).

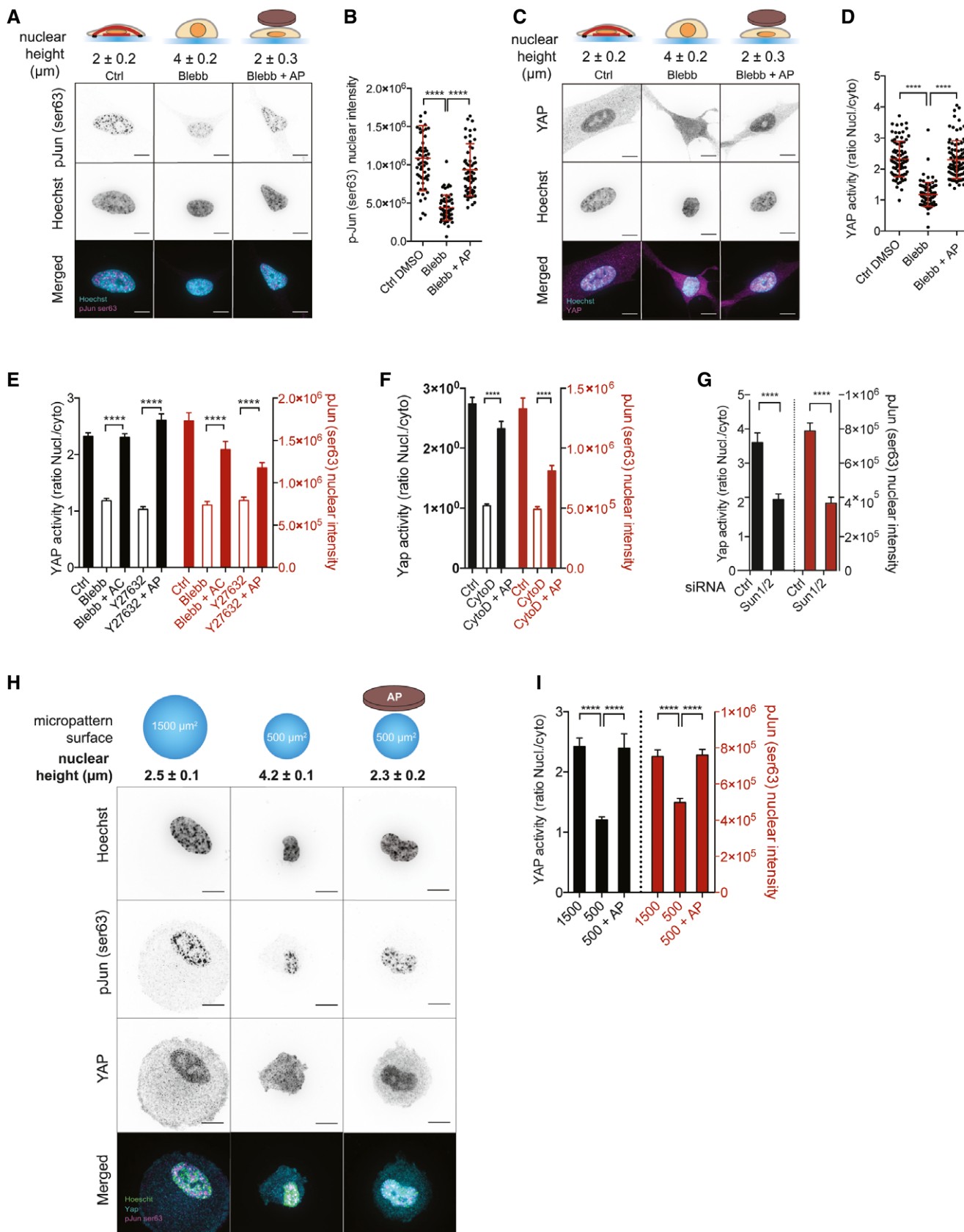

**Figure 3.**

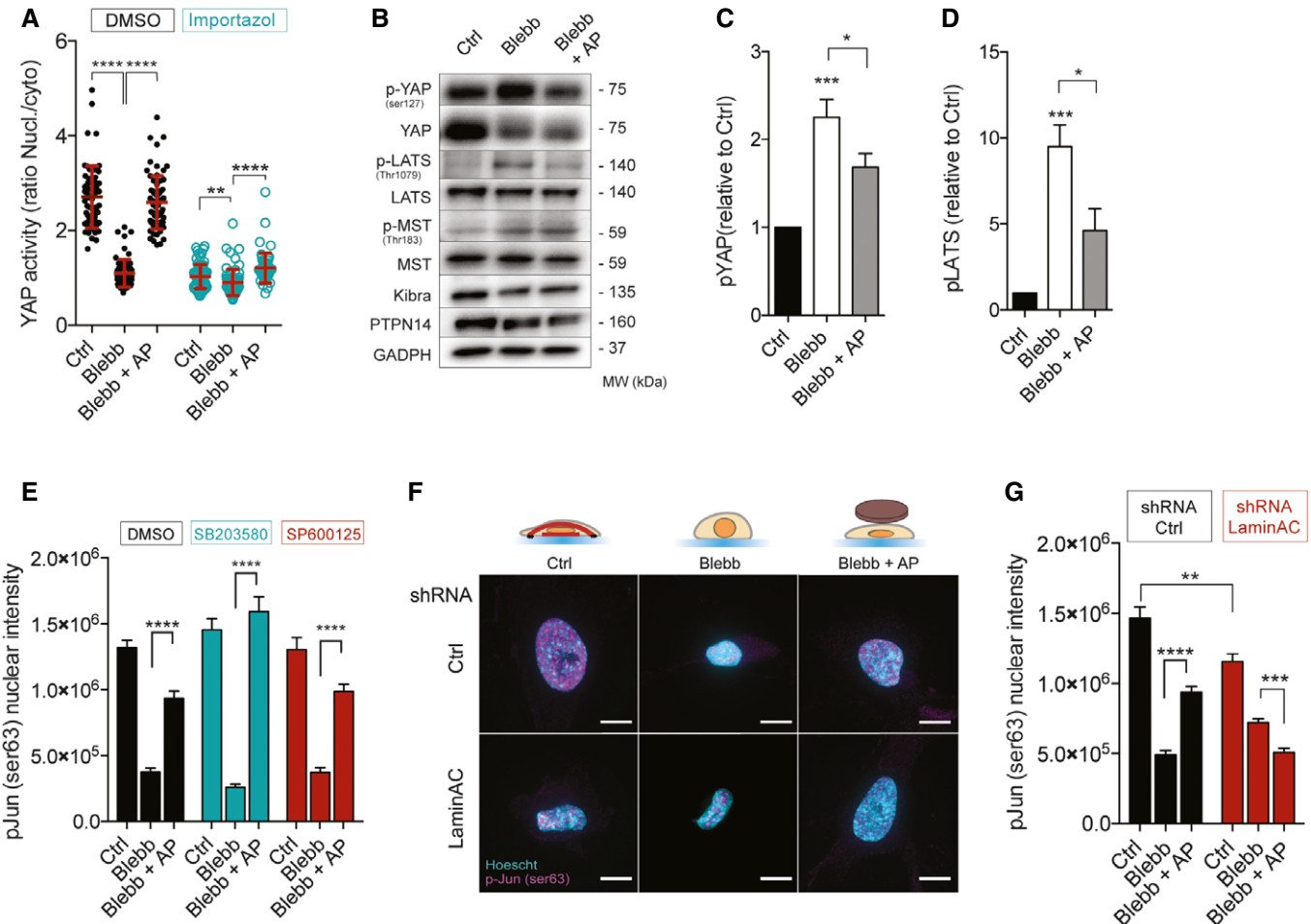

**Figure 4. Activation of TEAD and AP1 is distinctively controlled by nuclear shape.**

A   Quantifications of YAP activity in Ctrl, Blebb, and Blebb + AP conditions, treated or not with importazole (nucleo-cytoplasmic ratio). Data are presented as mean ± s.e.m. ($n$ = 59 minimum from three independent experiments, **$P$ < 0.05, ****$P$ < 0.001 one-way ANOVA—Tukey's multiple comparisons post-test).

B   Immunoblots of p-YAP (Ser127), YAP, p-LATS (Thr1079), LATS, p-MST (Thr1083), MST, Kibra, PTPN14, and GADPH for Ctrl, Blebb, and Blebb + AP conditions.

C   Corresponding quantification of p-YAP (Ser125) relative to Ctrl and normalized to GAPDH. Data are presented as mean ± s.e.m. ($n$ = 8, *$P$ < 0.5, ***$P$ < 0.01 one-way ANOVA—Tukey's multiple comparisons post-test).

D   Corresponding quantification of p-LATS (thr1079) relative to Ctrl and normalized to GAPDH. Data are presented as mean ± s.e.m. ($n$ = 6, *$P$ < 0.5, ***$P$ < 0.01 one-way ANOVA—Tukey's multiple comparisons post-test).

E   Quantifications of p-Jun Ser63 nuclear intensity in Ctrl, Blebb, and Blebb + AP conditions for cells treated with DMSO, SB203580, a MAP Kinase inhibitor, and SP600125, a JNK inhibitor. Data are presented as mean ± s.e.m. ($n$ = 18 minimum, ****$P$ < 0.001 one-way ANOVA—Tukey's multiple comparisons post-test.).

F   Representative cells stained for p-Jun Ser63 (magenta) and for DNA (cyan) in Ctrl, Blebb, and Blebb + AP conditions for cells depleted or not for lamin A/C. Scale bar = 10 μm.

G   Corresponding quantification of p-Jun Ser63 nuclear intensity. Data are presented as mean ± s.e.m. ($n$ = 49 minimum, **$P$ < 0.05, ***$P$ < 0.01, ****$P$ < 0.001 one-way ANOVA—Tukey's multiple comparisons post-test).

We next addressed the molecular mechanism of c-Jun activation by nuclear flattening. c-Jun is phosphorylated by Jun N-terminal kinase (JNK) and p38 mitogen-activated protein kinase family members. However, pharmacological inhibition of JNK or p38 did not affect c-Jun phosphorylation on Ser63 in response to nuclear flattening (Fig 4E). We then focused on other NE proteins, which have been reported as mechanosensitive, such as ATR [37] and lamin A/C [10,18]. While ATR pharmacological inhibition had no effect on YAP activation or c-Jun phosphorylation (Fig EV5H), we found that depletion of lamin A/C completely prevented c-Jun

phosphorylation in response to nuclear flattening (Figs 4F and G, and EV5F). Lamin A/C depletion did not have any effect on YAP localization (Fig EV5G), suggesting that nuclear flattening activates YAP and c-Jun via partly independent pathways. Studies have showed that, in response to serum, lamin A/C interact with AP1 [38,39] and release it to allow its rapid activation [38]. As previously described [38], the lamina may serve as a scaffold that promotes AP1 interaction with its upstream kinase and/or phosphatase. Many serine/threonine kinase has been found associated with the NE [40] and may participate to c-Jun activation independently of JNK.

## Discussion

The shape of membrane-bound organelle, such as mitochondria, can influence a variety of cellular processes [41]. While it was shown that tension within the nuclear envelope can regulate transcription [17,20,42], our study allowed the identification of four transcription factors (out of 345 tested) which are sensitive to nuclear flattening. Among them is TEAD, whose upstream regulator YAP has been recently shown to be activated by compressive force applied on the nucleus via NPC stretch [20]. Myosin-dependent contractility or externally applied mechanical tension is known to regulate gene expression [21,43] and cell cycle progression [23,24] but we show here that nuclear flattening in the absence of myosin II contractility is sufficient to activate TEAD and AP1. While we have given a number of arguments demonstrating the causal role of NE deformation in activating TEAD and AP1, we cannot rule out the possibility that cytoplasmic mechanotransduction pathways may also contribute to regulate their activity in response to cellular deformation. Interestingly, the mechanisms that mediate TEAD and AP1 activation in response to nuclear flattening appear to be independent, but both occur during G1 and promote G1/S transition. While the molecular mechanisms mediating G1 nuclear flattening remain to be identified, it may result from the increasing contractility developed by cells during G1 [44] and/or by the associated adhesion remodeling [45].

Whereas NE fluctuations were reported during cell cycle progression, their consequences on cell growth are unclear [46]. Our results suggest that the molecular mechanisms that determine nuclear shape in proliferative tissue may play an important role during developmental and pathological processes and it is tempting to speculate that additional epigenetic mechanisms [42,47] may also be regulated by the nuclear morphology. During solid tumor development, the shape of the cell nucleus displays significant alterations, whose extent is used by cytopathologists to diagnose, stage, and prognose cancer [12]. Whereas altered nuclear morphology is a predictor of cancer aggressiveness and clinical outcome, it is still unclear how these changes in nuclear shape can affect cell behavior and gene expression. Our demonstration that nuclear shape-dependent regulation of cell cycle via AP1 and TEAD may contribute to cancer growth and it will be interesting in the future to determine whether AP1 and TEAD activity correlates with nuclear shape and cancer aggressiveness.

## Materials and Methods

### Cell lines, reagents, and antibodies

MRC-5 (ATCC®CCL-171™), HeLa (ATCC®CCL-2TM), and CCD-18Co cells were bought from ATCC and grown in Dulbecco's modified Eagle's medium (DMEM; Gibco) supplemented with 10% fetal bovine serum (Sigma) and antibiotic–antimycotic solution (Sigma). HeLa-FUCCI cells were a gift from Y. USSON (TIMC-IMAG Lab, Grenoble, France). Hydrogels with different stiffnesses coated with collagen were purchased from Matrigen. Cells were lysed directly in Laemmli buffer for Western blot (Tris–HCl pH 6.8 0.12 M, glycerol 10%, sodium dodecyl sulfate 5%, β-mercaptoethanol 2.5%, bromophenol blue 0.005%). Cells

were treated with different drugs: blebbistatin (Tocris) at 100 μM, importazole (Sigma) at 40 μM, okadaic acid (Santa Cruz) at 100 nM, SB203580 (Cell Signaling) at 25 μM, Y27632 (Tocris) at 50 μM, SP600125 (Sigma) at 50 μM, and BAPTA/AM (Sigma) at 50 μM. Cell viability was assed based on membrane integrity and esterase activity using staining with calcein/AM and ethidium homodimer (LIVE/DEAD™ Viability Kit—Thermofisher L3224). Antibodies are detailed in Table EV1.

### Cell synchronization

Cells were arrested in prometaphase using S-trityl-L-cysteine (STC) (Sigma-Aldrich) (5 μM; 16 h) and harvested by mitotic shake-off (Fig 1B–D). Cells were washed, then seeded on fibronectin-coated substrate, fixed at different step of the cell cycle, and cdt1 expression was assessed using immunofluorescence.

### Agarose pad and nuclear compression

To compress the nuclei, we adapted a method previously described by Dumont and Mitchison [25] as follows: A solution of 2% ultra-pure agarose (FMC Bio-Products) in serum-free DMEM medium was prepared and brought to boil, and 5 ml was poured in a 60-mm Petri dish to solidify. A circular pad area (33 mm diameter) was cut out and placed in PBS buffer. Then, the agarose pad (AP) was deposited gently on cells and a 32-mm glass coverslip surmounted of a 125 mg washer was carefully placed above to maintain the cushion for compression.

### Plasmid

pEGFP C3-LATS1 was a gift from Marius Sudol (Addgene plasmid #19053). The point mutations T1079D and T1079E were introduced into wild-type pEGFP C3-LATS1 using Q5 site-directed mutagenesis kit (NEB). All constructs were verified by DNA sequencing. Phosphomimetic function of LATS1 was analyzed by Western blot analysis of p-YAP. Plasmids were transfected using Lipofectamine® 2000 (Invitrogen) according to the manufacturer's instructions. pRL-SV40P was a gift from Ron Prywes (Addgene plasmid # 27163), and 8xGTIIC-luciferase was a gift from Stefano Piccolo (Addgene plasmid # 34615). For YAP activity measurement, HeLa cells were co-transfected with 8xGTIIC-luciferase and pRL-SV40P and YAP activity was assessed using Dual-Lciferase Reporter Assay (Promega Ref E910) with Renilla luciferase as internal control. Activities were displayed as arbitrary units. GFP reporter for AP1 (Cignal AP1 Reporter Assay Kit GFP) was purchased from QIAGEN (CCS-011G). pLL3.7 K122-ires-GFP-TEAD-responsive-H2B mCherry reporter was a gift from Yutaka Hata (Addgene plasmid # 68714).

### Transcription factor activity array

The Affymetrix protein–DNA combo array kit was used according to the manufacturer's instructions. Briefly, this assay is designed to simultaneously analyze the binding activities of 345 transcription factors to their corresponding consensus DNA sequences. Each spot corresponds to a specific consensus DNA sequences. The nuclear extract samples from MRC-5 cells treated with

DMSO, Blebb, and Blebb + AP were mixed with pre-labeled transcription factor probes. Each nuclear fraction was previously checked for their purity by Western blot. Probe mix (10 μl) was combined with 15 μg of nuclear extract for each condition. The array membrane was exposed using ChemiDoc apparatus (Bio-Rad). The intensity of each spot (Fig EV1F), representing the binding activity of transcription factors to consensus DNA sequences, was measured with ImageJ software and presented as an arbitrary value. Overexposed signals were discarded, and minimum fold change threshold was applied (Ctrl versus Blebb and Blebb + AP versus Blebb). Selected TFs (shown in Figs 1D and EV1E) are those whose activity level in compressed nuclei (Blebb + AP) was similar to control cells (between 0.5- and 1.5-fold). AP1 activity in Figs 1 and EV3A was analyzed using the AP1 Reporter Kit from BPS Bioscience.

## Microscopy

Cells were fixed with 3.7% PFA (Sigma) for 20 min, permeabilized with 0.1% Triton in PBS (Pigma), then washed with PBS, and blocked with a blocking solution (2.5% bovine serum albumin in PBS Tween 0.2%) for 1 h. Samples were incubated overnight at 4°C with primary antibody in blocking solution, followed by three washes with PBS Tween 0.2%. The cells were then incubated with secondary antibody at room temperature for 1 h followed by three washes with PBS Tween 0.2%. After 2 washes with PBS, samples were finally mounted using mounting medium with DAPI (Prolong—Invitrogen). Samples were observed using a Spinning Disk Andromeda RILL-FEI (EMCCD iXon 897 Camera with alpha-Plan Apo 63×/1.46 oil objective). Fields were randomly imaged, and areas of interest were analyzed after Z stack projection (Z project-maximum intensity). For p-Jun (Ser63 and Ser73) fluorescent intensity analysis, corrected total nuclear fluorescence (CTNF) was calculated following this formula: CTNF = Integrated Density – (Area of nucleus × mean florescence of background readings). Adhesions, cells viability, and FUCCI cell cycle were imaged using epifluorescence microscopy with a Plan-Apochromat 40×/1.4 Oil objective (for adhesions), with a Plan-Neofluar 20×/0.5 (for HeLa-FUCCI), and with a Plan-Neofluar 10×/0.3 (for cell viability). Adhesion size and number were analyzed using the Focal Adhesion Analysis Server (http://faas.bme.unc.edu/). All pictures were analyzed using FIJI®.

## Micropattern

Fibronectin-patterned glass coverslips were microfabricated using the first steps of the glass technique described by Vignaud *et al* [48]. Briefly, glass coverslips (VWR) were plasma treated for 30 s and incubated for 30 min at room temperature (RT) with 0.1 mg/ml poly-L-lysine-grafted-polyethylene glycol (pLL-PEG, SuSoS) diluted in HEPES (10 mM, pH 7.4, Sigma). After washing in deionized phosphate-buffered saline (dPBS, Life technologies), the pLL-PEG covered coverslip was placed with the polymer brush facing downwards onto the chrome side of a quartz photomask (Toppan) for photolithography treatment (5-min UV-light exposure, UVO Cleaner Jelight). Subsequently, the coverslip was removed from the mask and coated with 20 μl/ml fibronectin (Sigma) diluted in dPBS for 30 min at RT.

## shRNA

Lentiviral shRNA targeting human SUN1, SUN2, lamin A/C, and lentiviral non-targeting control vector were purchased from Open Biosystems.

Lamin A/C shRNA #1 (Oligo ID: TRCN0000061836) hairpin sequence: 5′-CCGGCATGGGCAATTGGCAGATCAACTCGAGTTGATCTGCCAATTGCCCATGTTTTTG-3′,
Lamin A/C shRNA#2 (Oligo ID TRCN0000061837) hairpin sequence: 5′-CCGGGCCGTGCTTCCTCTCACTCATCTCGAGATGAGTGAGAGGAAGCACGGCTTTTTG-3′
SUN1 shRNA#1 (Oligo ID TRCN0000133901) hairpin sequence: 5′-CCGGCAGATACACTGCATCATCTTTCTCGAGAAAGATGATGCAGTGTATCTGTTTTTTG-3′
SUN2 shRNA#1 (OligoIDTRCN0000141958) hairpin sequence: 5′-CCGGGCAAGACTCAGAAGACCTCTTCTCGAGAAGAGGTCTTCTGAGTCTTGCTTTTTTG-3'

## siRNA

Cells were transfected using Lipofectamine® RNAiMAX (Invitrogen) according to the manufacturer's instructions with the corresponding siRNA: SilencerTM Negative Control No. 1 siRNA (AmbionTM AM4611); siRNA targeting: Jun (human) was purchased from Life technologies (Silencer® Select s7659 and s7660); YAP (human) was purchased from Life technologies (Silencer® Select s534572 and s20367).

## Atomic force microscopy

An AFM (CellHesion module; JPK Instruments) mounted on an optical microscope (Olympus) was used to perform single cell height measurement (Fig EV1H) and nuclear compression (Fig EV2). Experiments were conducted at 37°C using the Petri Dish Heater system and DMEM medium buffered with 20 mM HEPES and complemented with 1% FBS. Tipless cantilevers (arrow-TL1-50) with a nominal force constant of 0.03 N/m were used and calibrated using the thermal noise method. To measure cell height, MRC-5 was cultured on soft (1 kPa) and stiff (50 kPa) collagen-coated substrate for 24 h. Following an approach on the adjacent surface, the cantilever was positioned over the cell to measure the height delta. For nuclear compression (Fig EV2), cells were plated on 35-mm dish from ibidi (gridded surface). After positioning the cantilever above the nucleus, constant height mode was used (target height of 3 μm below the surface) and maintained for 120 min.

## Real-time PCR with reverse transcription

Total RNA was purified from cells using the RNAqueous-Micro kit (Ambion-Life technologies) according to the manufacturer's instructions. RNAs were reverse-transcribed into cDNA using the High-Capacity cDNA Reverse Transcription Kit (Applied Biosystems). High-throughput real-time qPCR was performed by the Center for Gastrointestinal Biology and Disease Advanced Analytics (AA) Core at UNC (Chapel Hill NC 27599). The 48 TaqMan primer references are listed in Table EV2. Expression data were normalized to a standard curve generated from a pool of control cells. GAPDH was used

as the reference gene. Data are based on results from minimum four independent experiments. Only genes with increase in mRNA expression superior to 1.5-fold change in response to nuclear flattening were selected.

### Flow cytometry

To assess the proliferative activity, 5-ethynyl-2′-deoxyuridine (EdU) was added to cells just before applying the agarose cushion for 2.5 h. Cells were stained, as described in the Click-iT EdU Alexa Fluor 647 Imaging Kit. To determine the proliferative indices of cells, FACS analysis (Sony SH800) was performed according to the manufacturer's instructions. A total of at least 60,000 events per condition for three independent experiments were recorded for the analysis.

### Protein–Protein Interaction Network reconstruction and analysis

Literature-based protein–protein interaction involving AP1, PPARA, TEAD1, SP1, and mechano-related proteins was gathered from BIOGRID database (http://thebiogrid.org/, release archive 3.4.156; [49]) with PSIQUIC retrieval. The resulting interaction Table (Table EV3) was curated to represent only the interactions validated in *Homo sapiens*, and the network was generated in cytoscape (http://www.cytoscape.org/; Shannon *et al, 2003*). For visualization purpose, the self-loops and multiple edges were removed. A solution of a minimal network that includes the transcription factors was determined by iteration of shortest path analysis and network parameter analysis-based pruning. The custom list of mechano-related proteins (Tables EV3 and EV4) was built over an assembly of keyword indexed proteins in UniProt (goa: "mechanical") and GO terms in QuickGO databases (GO:0050982, "Detection of mechanical stimulus"; GO:0071260, "Cellular response to mechanical stimulus"; GO:0009612, "Response to mechanical stimulus"). Upon request, generated network maps can be uploaded for public access on CyNetShare (www.cynetshare.ucsd.edu).

### Statistics

Statistical analysis was performed using GraphPad Software. Data are presented as mean ± s.e.m. Unpaired *t*-test has been used unless stated otherwise. Besides for transcription factor activity analysis (as described above), no exclusion criteria were used. The numbers of independent experiments performed for all of the quantitative data are indicated in the Figure legends.

Expanded View for this article is available online.

### Acknowledgements
The authors thank the cell imaging facility MicroCell and its outstanding staff, including Alexei Grichine, Mylène Pezet, and Jacques Mazzega for their technical assistance. We thank Keith Burridge for his continuous support. C.G. is supported by grants from the Agence National de la Recherche (ANR-13-JSV1-0008) and from European Research Council (ERC) under European Union's Horizon 2020 research and innovation program (ERC Starting Grant n_639300). The authors thank the Center for Gastrointestinal Biology and Disease (CGIBD) Advanced Analytics (AA) Core (NIH P30 DK034987) at the University of North Carolina (Chapel Hill, NC). L.V.L. is supported by grants from NCSU CVM (Seed Funding), UNC CGIBD (Pilot/Feasibility Grant NIH P30 DK034987) and from the University of North Carolina Lineberger Comprehensive Cancer Center (Developmental grant).

### Author contributions
JA and CG designed experiments. JA performed experiments and analyzed data. VB-R, LP, BEH, and SF helped with experimental design and procedures. MB and TA designed and fabricated the micropatterned surfaces. CB performed protein–protein interaction bioinformatic analysis. GB and LVL designed and performed flow cytometry analysis. CG directed the project and wrote the manuscript. All authors provided detailed comments.

### Conflict of interest
The authors declare that they have no conflict of interest.

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
