## [Review Process File · EMBO Reports]

Nuclear envelope deformation controls cell cycle progression in response to mechanical force

Julien Aureille, Valentin Buffière-Ribot, Ben E. Harvey, Cyril Boyault, Lydia Pernet, Tomas Andersen, Gregory Bacola, Martial Balland, Sandrine Fraboulet, Laurianne Van Landeghem and Christophe Guilluy

Review timeline:	Submission date:	13 March 2019
	Editorial Decision:	14 March 2019
	Revision received:	13 May 2019
	Editorial Decision:	11 June 2019
	Revision received:	18 June 2019
	Accepted:	3 July 2019

Editor: Deniz Senyilmaz-Tiebe

Transaction Report: This manuscript was transferred to *EMBO reports* following peer review at *The EMBO Journal*

1st Editorial Decision

14 March 2019

Thank you for submitting your manuscript entitled "Nuclear envelope deformation controls cell cycle progression in response to mechanical force" to EMBO Reports. Your manuscript was previously reviewed at another journal, and you transferred your manuscript to EMBO Reports along with the referee reports. I have now looked at everything carefully.

Given the interest of the referees on the study I would like to invite you to revise your manuscript with the understanding that the referee comments must be fully addressed and their suggestions taken on board. Referee #1 finds that functional relevance of the findings is not sufficiently explored due to usage of a single cell line. Moreover referees point out that as it stands the experimental approach does not allow differentiating the effects of nuclear deformation and cellular deformation (2nd concern of referee #1 and 1st concern of referee #2).

Please address all referee concerns in a complete point-by-point response. Acceptance of the manuscript will depend on a positive outcome of the review. It is EMBO Reports policy to allow a single round of revision only and acceptance or rejection of the manuscript will therefore depend on the completeness of your responses included in the next, final version of the manuscript.

You can submit the revision either as a Scientific Report or as a Research Article. For Scientific

Reports, the revised manuscript can contain up to 5 main figures and 5 Expanded View figures. If the revision leads to a manuscript with more than 5 main figures it will be published as a Research Article. If a Scientific Report is submitted, these sections have to be combined. This will help to shorten the manuscript text by eliminating some redundancy that is inevitable when discussing the same experiments twice. In either case, all materials and methods should be included in the main manuscript file.

Supplementary/additional data: The Expanded View format, which will be displayed in the main HTML of the paper in a collapsible format, has replaced the Supplementary information. You can submit up to 5 images as Expanded View. Please follow the nomenclature Figure EV1, Figure EV2 etc. The figure legend for these should be included in the main manuscript document file in a section called Expanded View Figure Legends after the main Figure Legends section. Additional Supplementary material should be supplied as a single pdf labeled Appendix. The Appendix includes a table of content on the first page with page numbers, all figures and their legends. Please follow the nomenclature Appendix Figure Sx throughout the text and also label the figures according to this nomenclature. For more details please refer to our guide to authors.

When preparing your letter of response to the referees' comments, please bear in mind that this will form part of the Review Process File, and will therefore be available online to the community. For more details on our Transparent Editorial Process, please visit our website: http://emboj.embopress.org/about#Transparent_Process

Regarding data quantification, please ensure to specify the name of the statistical test used to generate error bars and P values, the number (n) of independent experiments underlying each data point (not replicate measures of one sample), and the test used to calculate p-values in each figure legend. Discussion of statistical methodology can be reported in the materials and methods section, but figure legends should contain a basic description of n, P and the test applied. Please also include scale bars in all microscopy images.

We now strongly encourage the publication of original source data with the aim of making primary data more accessible and transparent to the reader. The source data will be published in a separate source data file online along with the accepted manuscript and will be linked to the relevant figure. If you would like to use this opportunity, please submit the source data (for example scans of entire gels or blots, data points of graphs in an excel sheet, additional images, etc.) of your key experiments together with the revised manuscript. Please include size markers for scans of entire gels, label the scans with figure and panel number, and send one PDF file per figure.

- a complete author checklist, which you can download from our author guidelines (<http://emboj.embopress.org/authorguide#revision>). Please insert page numbers in the checklist to indicate where the requested information can be found.
 - a letter detailing your responses to the referee comments in Word format (.doc)
 - a Microsoft Word file (.doc) of the revised manuscript text
 - editable TIFF or EPS-formatted figure files in high resolution
- (In order to avoid delays later in the publication process please check our figure guidelines before preparing the figures for your manuscript:
http://www.embopress.org/sites/default/files/EMBOPress_Figure_Guidelines_061115.pdf)
- a separate PDF file of any Supplementary information (in its final format)
 - all corresponding authors are required to provide an ORCID ID for their name. Please find instructions on how to link your ORCID ID to your account in our manuscript tracking system in our Author guidelines (<http://emboj.embopress.org/authorguide>).

As part of the EMBO publication's Transparent Editorial Process, EMBO reports publishes online a Review Process File to accompany accepted manuscripts. This File will be published in conjunction with your paper and will include the referee reports, your point-by-point response and all pertinent correspondence relating to the manuscript.

I look forward to seeing a revised version of your manuscript when it is ready. Please let me know if you have questions or comments regarding the revision.

REFeree REPORTS

Referee #1

In the MS, the authors describe a mechanism whereby nuclear flattening in HeLa cells triggers YAP/TEAD and AP-1 activation, thus promoting S-phase entry. Nuclear flattening induces YAP nuclear accumulation (with a phosphorylation-dependent, but Hippo-independent mechanism) and Jun phosphorylation (with a mechanism entailing lamin A/C). Together activated YAP and Jun promote the transcription of growth-promoting genes (as previously reported).

GENERAL COMMENT

It was recently reported that force application to the nucleus is sufficient to induce YAP nuclear relocalization, and that YAP nuclear entry requires LINC-mediated mechanical coupling between the cytoskeleton and the nucleoskeleton (Elosegui-Artola 2017). Here the authors add a functional layer, showing YAP transcriptional responses, and YAP+JUN dependent DNA synthesis in blebbistatin-treated cells undergoing compression with and agarose pad, which was not reported in the previous work. While important, this can hardly be considered a substantial advancement for the field. Moreover, it remains unclear what the physiological counterpart of this in vitro experimental setup might be.

Mechanical activation of AP-1 transcription factors was previously reported in several cellular contexts, as well. Here the authors propose that the mechanism entails Jun phosphorylation, which would depend on the nuclear lamina but not on JNK. This mechanism is interesting, but really too preliminary. We have just scattered pieces of evidence and a lot of inference. Also, the bioinformatic search for mechanoregulators that may impact on YAP and AP-1 is interesting, but unluckily it has no experimental validation.

The MS contains some interesting hints to novel processes, but in its current form the novelty, and data supporting novel elements, are too scarce to recommend the publication in this Journal.

MORE SPECIFIC POINTS (although addressing them, in no case, in my opinion, will be able to overcome the above limitations)

All experiments are performed in one cell line, leaving several open issues, such as: does nuclear flattening at the G1/S transition happen in other replicating cells? (And beyond this: what could happen in a tissue, rather than in cultured cells?)

Most experiments use large agarose pads that induce a cellular deformation. Yet, the authors make very specific claims about the causal role of nuclear deformation in transcriptional and proliferative responses in HeLa cells. Simultaneous loss-of-function approaches targeting the LINC complex or of LaminA/C should be used, to demonstrate the relevance of nuclear mechanotransduction downstream of cell compression. Also, only most experiments are performed only with blebbistatin, in particular those investigating transcriptional and biological effects; some of these experiments should be performed with other inhibitors or low physical stimulation of cells (as the authors did for biochemical/localization assays in figure 4).

More than one siRNA/shRNA sequence should be used for each target.

Referee #2

In this article, Aureille and colleagues perform a very elegant experiment in which they show that simply deforming cells in which Myosin II is inhibited by blebbistatin, can rescue their cell cycle progression. They attribute this effect to the deformation of the nucleus, which is lost when Myosin II is inhibited and rescued by confinement. While this outcome could have been predicted from previous work on nuclear deformation and YAP translocation, the experiment was actually never reported before, at least to my knowledge. The authors then use classical tools to explore the transcriptional pathways rescued by confinement in myo II inhibited cells. What they find is not unexpected, especially the fact that the YAP/TEAD pathway is rescued by cell confinement (also confirming their interpretation that the effect is likely due to nuclear deformation), but it is clearly shown and might open the way to further investigation of the effect of nuclear deformation on cell cycle progression. This has been postulated before, but never shown so directly. I think the article clearly deserves publication and will interest a broad readership. My main concern is not about the results, which are all very nice and well documented, but more about the writing, which is overinterpreting the results and not giving enough credit to prior knowledge.

Main concerns:

- The authors rule out the effect of direct classical mechanotransduction via focal adhesions upon confinement, while this is clearly another obvious pathway affected by Myosin II inhibition and potentially rescued by confinement.

It is thus important to provide enough evidence for that statement. They only show that focal adhesions do not change in size. This is not enough to be convincing. They could use a number of alternative strategies, like plating cells on non adhesive substrates to show that nuclear deformation is still enough to activate the pathways they revealed. They could also use inhibitors of the focal adhesion signaling pathways. Or they could rephrase their conclusions to leave open the possibility that these pathways also contribute to the effect they observe.

- A first overstatement is made from the conclusion of the TFs study: 'Our results reveal that nuclear flattening...indicating that nuclear shape is a major regulator of their activity'. I think what is shown is certainly not enough to prove that nuclear shape is a major regulator of the activity of these promoters. First the direct causality between nuclear shape and activation is not proven here, second the fact that it would be a 'major' regulator is not even addressed at all. The authors need to find the right sentence that summarised their findings.

- The study they perform to link the TFs they identified to response to mechanical stress is also a very weak point to prove that 'nuclear morphology contributes to these (mechanical stress) pathways'. I think it is so weak that it could totally be removed from the article.

- There is also a problem with the positioning of the article relative to the recent finding from Roca-Cusachs lab that shows that, even in absence of actin, myosin and LINC complex, YAP can be induced to enter the nucleus by simple deformation of the nucleus. The current article is basically verifying a straightforward prediction of the Roca-Cusachs paper, published one year ago in *Cell*. In my opinion, instead of commenting on that paper as if it was not a major prior knowledge for the current study, they should really already discuss it in the introduction and base the rationale for their study on the results of that paper. This will make more clear what their study brings. And I think it still brings very important additional insights - it shows that this prediction of the effect of nuclear shape on cell cycle progression, which seems to be a natural consequence of the previous knowledge, is really true, and this is important, and it also provides a more comprehensive study of the pathways involved.

- The authors should also discuss the work of Marco Foiani, published in *Cell* in 2014, showing that nuclear deformation induces recruitment of activation of ATR at the nuclear envelope. This result provides a quite obvious candidate to test in their study, in addition to the YAP pathway. The authors mention the phosphorylation of c-Jun in response to nuclear flattening. Could it be dependent on ATR? Even if that particular question might be beyond the scope of the paper, ATR needs to be discussed (also because there are only very few pathways which have been shown to be directly related to nuclear or nuclear envelope deformation).

- The last part of the results, including the effect of confinement of cells on small patterns and the rescue of Y27632 treatment seem to rather belong to the beginning of the paper rather than the end.

It is not obvious that it brings really something more than what was shown from the rescue of Myosin II treatment. It shows the same thing, with additional treatments, and it is thus an interesting point to make at the beginning to reinforce the initial conclusion made from the Myosin II inhibition experiments. Placed at the end, it gives a strange inconclusive end to the paper.

- The beginning of the discussion contains a number of statements which are not correct: 1) It is not clear how their study, more than the previous ones, define the nucleus as a 'sensor' that is triggered by mechanical tension. This could be concluded from the previous studies on consequences of nuclear deformation. But showing that its deformation induces cellular responses, is not enough to define it as a 'sensor' in my opinion. A sensor of what? In the context of a tissue what would be the signal sensed by nuclear deformation to instruct cell cycle progression?

A second point is not exact: 'The impact of nuclear shape on gene expression was not known'.

Maybe it was still not clear, and it is still not even after this paper. But not 'not known'!!!

- The authors could also discuss the recent article from X. Trepap lab on the regulation of G1 length (and thus G1/S transition) by traction forces (NCB 2018). This article is not related to nuclear deformation but shows that a major regulator of cell cycle progression is the total energy developed by cells as they pull on the substrate. So it is totally compatible with the conclusions shown here.

- Another relevant article was published in PNAS in 2017 by the lab of Zydovska and shows that as cells transit from G1 to S/G2, their nuclei get more 'tensed' and show reduced nuclear envelope fluctuations. They do not show that this regulates G1/S transition, but again, it is in good agreement with the mechanism proposed here by the authors, that nuclear envelope tension and cell cycle progression are related.

Minor points:

- I am not sure I understood that point very well, but it seems that of the 17 TFs inhibited by blebbistatin, 15 are rescued by confinement! This is a lot! And suggests also that the two that are not have maybe something very special and interesting.

Referee #3

In this manuscript, Aureille et al. report that nuclear deformation, and more specifically its flattening, controls cell cycle progression. By employing an elegant setup to compress cells in the vertical direction and flatten nuclei, the authors first show that nuclear flattening specifically induces proliferation by triggering G1 to S transition. Then, the authors identify the patterns in transcription factor activity associated, and establish that the transcription factors TEAD and AP1 are specifically activated. Subsequently, they explore the mechanisms by which this occurs, which are different for each transcription factor. Finally, the authors identify that nuclear flattening mediates the known effects of ROC-mediated contractility, or cell spreading. The results of this manuscript reveal an important modulation of cell cycle progression by a purely physical parameter, nuclear shape, which is of great interest and may have potential major implications in several physiological scenarios. The work is carried out in an elegant, systematic, and rigorous way, and I therefore strongly support publication. I only have a set of minor comments to be addressed before publication.

1. My main comment is that even though the authors explore different mechanisms for the activation of YAP and c-Jun according to previously reported possibilities, it would be interesting to check the respective perturbations with the other candidate as well. That is, does lamin A/C depletion also affect YAP levels, and does importazole also somehow affect c-JUN phosphorylation? This would help to discriminate to what extent the mechanisms are indeed independent.

2. I find it interesting that LATS mutations affect YAP ratios in control cells, but does not impact the effect of forced nuclear flattening. A potential explanation for this would be that LATS mutations somehow affect nuclear shape per se, and this impacts YAP nuclear ratios aside from a direct effect of the hippo pathway. This would explain why if this nuclear deformation is forced exogenously, LATS has no effect. The authors could measure nuclear shapes in LATS-mutated cells to assess this, this would in fact be very interesting since it would help reconcile conflicting reports on the role of the hippo pathway in YAP mechanoresponses (an aspect which the authors themselves mention in the text).

3. In their discussion, the authors mention that "the impact of nuclear shape on gene expression was not known". This is an overstatement, as there are several works that analyse this from different

perspectives (see for instance the work by the Shivashankar group in Singapore). This does not remove novelty from this very interesting work, but the authors should rephrase this to properly acknowledge previous work.

4. As a minor aspect, the authors mention that in the case of YAP, inhibition of nuclear import decreased YAP translocation, but that a significant effect of nuclear flattening remained. This is then used to argue that this may not fully explain the results. Whereas it is true that a significant effect remains, it is extremely small, and actually the YAP ratios with blocked import are all almost exactly in the range of blebbistatin-treated cells. In this case, I would thus say that the differences in conditions of import inhibition are significant, but not biologically relevant. Thus, the authors' results are fully consistent with a regulation by nuclear import as previously reported. Again, I don't think this reduces the novelty of the work in any way - the current work goes well beyond previous findings in many ways - but this should be clarified.

5. Minor: the authors state that the agarose pads employed do not lead to nuclear herniation, but that data is not shown. Please show.

1st Revision - authors' response

13 May 2019

Response to Reviewer #1 comments:

In the MS, the authors describe a mechanism whereby nuclear flattening in HeLa cells triggers YAP/TEAD and AP-1 activation, thus promoting S-phase entry. Nuclear flattening induces YAP nuclear accumulation (with a phosphorylation-dependent, but Hippo-independent mechanism) and Jun phosphorylation (with a mechanism entailing lamin A/C). Together activated YAP and Jun promote the transcription of growth-promoting genes (as previously reported).

Comment 1: All experiments are performed in one cell line, leaving several open issues, such as: does nuclear flattening at the G1/S transition happen in other replicating cells? (And beyond this: what could happen in a tissue, rather than in cultured cells?)

Our response: We analyzed nuclear height during cell cycle progression in three distinct cell lines (Hela Fucci, MRC5 cells and MEFs) and we observed a significant decrease in nuclear height over the course of G1 in every cell lines tested, suggesting that nuclear flattening before G1/S transition may occur in several adherent cells. These data have been included in the revised version (figure 1b-e). As reviewer 1 suggested, it would be interesting to monitor nuclear shape (and G1 duration) in dividing cells *in vivo*, although this would require substantial further investigations that are beyond the scope of this current work whose objective was to explore the effect of nuclear flattening on the transcriptional machinery.

Comment 2: Most experiments use large agarose pads that induce a cellular deformation. Yet, the authors make very specific claims about the causal role of nuclear deformation in transcriptional and proliferative responses in HeLa cells. Simultaneous loss-of-function approaches targeting the LINC complex or of LaminA/C should be used, to demonstrate the relevance of nuclear mechanotransduction downstream of cell compression. Also, only most experiments are performed only with blebbistatin, in particular those investigating transcriptional and biological effects; some of these experiments should be performed with other inhibitors or low physical stimulation of cells.

Our response: While we have a number of arguments supporting the causal role of nuclear deformation in triggering AP1 and TEAD activation (detailed below), we cannot rule out a potential involvement of overall cellular deformation in our experimental system. Therefore we rephrased our conclusion to leave open the possibility that cytoplasmic pathways may also contribute to the effect we observe, as suggested by reviewer 2 (2nd paragraph of the result section and discussion). Our main arguments supporting an involvement of nuclear deformation include experiments that were presented in the initial version of the manuscript and additional experiments that have been added in the revised version: **1) Agarose pad-induced nuclear flattening does not affect adhesion maturation.** We have extended our analysis of integrin-based adhesions in order to ensure that adhesion-mediated mechanotransduction pathways are not activated by the application of the agarose pad in our experimental set-up. In addition to the initial analysis of adhesion size (sup figure 1e), we analyzed adhesion protein phosphorylation by immunofluorescence (new sup figure 1f) and

paxillin phosphorylation by western blot (new sup figure 1g). These experiments show that there is no increase in adhesion size, nor adhesion protein phosphorylation following addition of the agarose pad and indicate that the mechanotransduction pathways associated with adhesion maturation are not involved in the transcriptional effect observed in response to agarose pad-induced nuclear flattening **2) *Integrin-based adhesions are not required for AP1 and TEAD activation in response to nuclear compression.*** We cultured cells on poly-L-lysine surfaces to prevent adhesion formation and we analyzed the activity of TEAD and AP1 in response to compressive forces applied specifically on the nucleus using AFM (thus limiting global cellular deformation). We observed that nuclear compression by AFM activates TEAD and AP1 in cells cultured on low adhesion system (sup figure 2), indicating that integrin-based adhesion are not required. In addition, we cultured cells on fibronectin-coated micro-patterned surface in order to maintain adhesion surface unchanged and minimize overall cellular deformation while the agarose pad is applied. Consistent with our AFM experiments, we observed that agarose pad-induced nuclear flattening of cells plated on micro-patterned surfaces is sufficient to activate YAP and AP1 (c-Jun phosphorylation) (figure 3 h-i) **3) *Preventing cytoskeleton-dependent nuclear compression inhibits AP1 and TEAD activity.*** As the reviewer suggested, we targeted the LINC complex (SUN1 and SUN2 depletion) and we analyzed the effect on AP1 and TEAD activity (new sup figure 3 h-i), downstream pro-proliferative target genes (new sup figure 3j) and upstream regulators (figure 3g). All these experiments indicate that increasing nuclear height by disrupting the LINC complex inhibits TEAD and AP1 activity and their downstream effect on cell cycle progression. These results corroborate the data we obtained using the agarose pad and show that cytoskeleton-mediated nuclear compression activates TEAD and AP1. **4) *NPC and lamina are necessary for AP1 and TEAD activation in response to compression by the agarose pad.*** Finally, we found that altering known mechanosensitive structures located at the nuclear envelope (lamin A/C depletion and/or nuclear pore complex inhibition,) prevents YAP and AP1 activation in response to compression by the agarose pad. These structures have been reported to be mechanosensitive (reference10, 18 and 20) suggesting that forces applied on the NE triggers TEAD and AP1 activation in our experimental set-up.

Additionally, we have used distinct drugs to alter the actomyosin cytoskeleton such as cytochalasin D (added to the revised version), blebbistatin or ROCK inhibitor (Y27632). Whereas all these treatments increased nuclear height, addition of the agarose pad was sufficient to activate transcriptional upstream regulators (figure 3e-f; sup figure 4d, g) in presence of these inhibitors, indicating that nuclear flattening activates AP1 and TEAD in the absence of actomyosin-dependent contractility.

Comment 3: More than one siRNA/shRNA sequence should be used for each target.

Our response: We have used two different siRNA or shRNA sequence for knockdown experiments (see Method section). These experiments have been added to the revised version of the manuscript, and include siRNA and shRNA targeting SUN1 and SUN2 (shRNA Figure 3g and siRNA sup 4d), shRNA targeting lamin A/C (figure 4f-g and sup 5 f-g) and siRNA targeting YAP and c-Jun (sup figure 3g).

Response to Reviewer #2 comments

In this article, Aureille and colleagues perform a very elegant experiment in which they show that simply deforming cells in which Myosin II is inhibited by blebbistatin, can rescue their cell cycle progression. They attribute this effect to the deformation of the nucleus, which is lost when Myosin II is inhibited and rescued by confinement. While this outcome could have been predicted from previous work on nuclear deformation and YAP translocation, the experiment was actually never reported before, at least to my knowledge. The authors then use classical tools to explore the transcriptional pathways rescued by confinement in myo II inhibited cells. What they find is not unexpected, especially the fact that the YAP/TEAD pathway is rescued by cell confinement (also confirming their interpretation that the effect is likely due to nuclear deformation), but it is clearly shown and might open the way to further investigation of the effect of nuclear deformation on cell cycle progression. This has been postulated before, but never shown so directly. I think the article clearly deserves publication and will interest a broad readership. My main concern is not about the results, which are all very nice and well documented, but more about the writing, which is overinterpreting the results and not giving enough credit to prior knowledge.

Comment 1: The authors rule out the effect of direct classical mechanotransduction via focal adhesions upon confinement, while this is clearly another obvious pathway affected by Myosin II inhibition and potentially rescued by confinement. It is thus important to provide enough evidence for that statement. They only show that focal adhesions do not change in size. This is not enough to be convincing. They could use a number of alternative strategies, like plating cells on non adhesive substrates to show that nuclear deformation is still enough to activate the pathways they revealed. They could also use inhibitors of the focal adhesion signaling pathways. Or they could rephrase their conclusions to leave open the possibility that these pathways also contribute to the effect they observe.

Our response: Whereas we have a number of arguments supporting the causal role of nuclear deformation in triggering AP1 and TEAD activation (arguments are detailed in the response to comment#2 from reviewer 1), we cannot rule out a potential involvement of overall cellular deformation in our experimental system. Therefore we rephrased our conclusion to leave open the possibility that cytoplasmic pathways may also contribute to the effect we observe, as suggested by the reviewer (2nd paragraph of the result section and discussion). Regarding the involvement of adhesions, we have extended our analysis of integrin-based adhesions in order to ensure that adhesion-mediated mechanotransduction pathways are not activated by the application of the agarose pad in our experimental set-up. In addition to the initial analysis of adhesion size (sup figure 1e), we analyzed adhesion protein phosphorylation by immunofluorescence (new sup figure 1f) and paxillin phosphorylation by western blot (new sup figure 1g). These experiments show that there is no increase in adhesion size, nor adhesion protein phosphorylation following addition of the agarose pad and indicate that the mechanotransduction pathways associated with adhesion maturation are not involved in the transcriptional effect observed in response to agarose pad-induced nuclear flattening. Additionally, we cultured cells on poly-L-lysine surfaces to prevent adhesion formation and we analyzed the activity of TEAD and AP1 in response to compressive forces applied specifically on the nucleus using AFM (thus limiting global cellular deformation). We observed that nuclear compression by AFM activates TEAD and AP1 in cells cultured on low adhesion system (sup figure 2), indicating that integrin-based adhesion are not required. Finally, we cultured cells on fibronectin-coated micro-patterned surface in order to maintain adhesion surface unchanged and minimize overall cellular deformation while the agarose pad is applied. Consistent with our AFM experiments, we observed that agarose pad-induced nuclear flattening of cells on micro-patterned surfaces is sufficient to activate YAP and AP1 (c-Jun phosphorylation) (figure 3 h-i)

Comment 2: A first overstatement is made from the conclusion of the TFs study: 'Our results reveal that nuclear flattening...indicating that nuclear shape is a major regulator of their activity'. I think what is shown is certainly not enough to prove that nuclear shape is a major regulator of the activity of these promoters. First the direct causality between nuclear shape and activation is not proven here, second the fact that it would be a 'major' regulator is not even addressed at all. The authors need to find the right sentence that summarised their findings.

Our response: We removed the end of the sentence (including the “*major regulator of their activity*”), leaving a conclusion that describes more closely the experimental results: “*Our results reveal that agarose pad-induced nuclear flattening is sufficient to activate AP1, TEAD, PPAR and SP1 in the absence of myosin II-dependent contractility. However we cannot rule out the possibility that cytoplasmic mechanotransduction pathways may also contribute to activate these TFs, considering the cellular deformation that occurs following agarose pad application.*”

Comment 3: The study they perform to link the TFs they identified to response to mechanical stress is also a very weak point to prove that 'nuclear morphology contributes to these (mechanical stress) pathways'. I think it is so weak that it could totally be removed from the article.

Our response: We moved this panel to the supplementary figures (sup figure 3d) and use it as a confirmation that the identified transcription factors have been associated with other proteins known to participate to the cellular response to mechanical stress.

Comment 4: There is also a problem with the positioning of the article relative to the recent finding from Roca-Cusachs lab that shows that, even in absence of actin, myosin and LINC complex, YAP can be induced to enter the nucleus by simple deformation of the nucleus. The current article is basically verifying a straightforward prediction of the Roca-Cusachs paper, published one year ago in Cell. In my opinion, instead of commenting on that paper as if it was not a

major prior knowledge for the current study, they should really already discuss it in the introduction and base the rationale for their study on the results of that paper. This will make more clear what their study brings. And I think it still brings very important additional insights - it shows that this prediction of the effect of nuclear shape on cell cycle progression, which seems to be a natural consequence of the previous knowledge, is really true, and this is important, and it also provides a more comprehensive study of the pathways involved.

Our response: We agree with the reviewer's comment and in the revised version of the manuscript we discuss the paper from Roca-Cusachs group and a paper from the Niethammer group in the introduction (as well as in the result section), as important previous knowledge showing that changes in nuclear morphology can affect transcription.

Comment 4: The authors should also discuss the work of Marco Foiani, published in Cell in 2014, showing that nuclear deformation induces recruitment of activation of ATR at the nuclear envelope. This result provides a quite obvious candidate to test in their study, in addition to the YAP pathway. The authors mention the phosphorylation of c-Jun in response to nuclear flattening. Could it be dependent on ATR? Even if that particular question might be beyond the scope of the paper, ATR needs to be discussed (also because there are only very few pathways which have been shown to be directly related to nuclear or nuclear envelope deformation).

Our response: We tested the effect of ATR inhibitor and observed that ATR inhibition does not prevent AP1 and TEAD activation in response to nuclear flattening. This result was added to the revised version of the paper (new sup figure 5h) as well as a reference to the paper from Marco Foiani.

Comment 5: The last part of the results, including the effect of confinement of cells on small patterns and the rescue of Y27632 treatment seem to rather belong to the beginning of the paper rather than the end. It is not obvious that it brings really something more than what was shown from the rescue of Myosin II treatment. It shows the same thing, with additional treatments, and it is thus an interesting point to make at the beginning to reinforce the initial conclusion made from the Myosin II inhibition experiments. Placed at the end, it gives a strange inconclusive end to the paper.

Our response: We thank the reviewer for his/her suggestion, which significantly improves the narrative of the result section. As suggested, these results appear earlier (figure 3e; 3h-i) in the revised version of the manuscript.

Comment 6: The beginning of the discussion contains a number of statements which are not correct: 1) It is not clear how their study, more than the previous ones, define the nucleus as a 'sensor' that is triggered by mechanical tension. This could be concluded from the previous studies on consequences of nuclear deformation. But showing that its deformation induces cellular responses, is not enough to define it as a 'sensor' in my opinion. A sensor of what? In the context of a tissue what would be the signal sensed by nuclear deformation to instruct cell cycle progression? A second point is not exact: 'The impact of nuclear shape on gene expression was not known'. Maybe it was still not clear, and it is still not even after this paper. But not 'not known'!!!

Our response: As reviewer 2 suggested we changed the text in the discussion. We replaced the first sentence (and removed the "sensor") by another sentence discussing that the NE may sense and respond to cellular deformation by regulating transcription factor activity. Regarding the second sentence, we replaced the sentence (and removed "was not known") and discuss previous studies reporting that changes in nuclear morphology can regulate transcription (including work from Niethammer and colleagues, Shivashankar's group and Roca-Cusachs group).

Comment 7: The authors could also discuss the recent article from X. Trepac lab on the regulation of G1 length (and thus G1/S transition) by traction forces (NCB 2018). This article is not related to nuclear deformation but shows that a major regulator of cell cycle progression is the total energy developed by cells as they pull on the substrate. So it is totally compatible with the conclusions shown here.

Our response: The demonstration that cell generated tension can regulate cell cycle progression was done before by Ingber's group (already cited in the initial version of the manuscript), but as the reviewer suggested we added a reference to the recent work from Trepac's group.

Comment 8: Another relevant article was published in PNAS in 2017 by the lab of Zydovska and shows that as cells transit from G1 to S/G2, their nuclei get more 'tensed' and show reduced nuclear envelope fluctuations. They do not show that this regulates G1/S transition, but again, it is in good

agreement with the mechanism proposed here by the authors, that nuclear envelope tension and cell cycle progression are related.

Our response: We discuss this study in the revised version of the discussion.

Minor points:

I am not sure I understood that point very well, but it seems that of the 17 TFs inhibited by blebbistatin, 15 are rescued by confinement! This is a lot! And suggests also that the two that are not have maybe something very special and interesting.

Our response: Out of 15 TFs identified during the screening, only 4 TFs showed functional activation (and transcriptional induction of the target genes). This may be due to additional transcriptional regulatory mechanisms, such as epigenetic silencing of the target promoter or coactivator requirement.

Response to Reviewer #3 comments

In this manuscript, Aureille et al. report that nuclear deformation, and more specifically its flattening, controls cell cycle progression. By employing an elegant setup to compress cells in the vertical direction and flatten nuclei, the authors first show that nuclear flattening specifically induces proliferation by triggering G1 to S transition. Then, the authors identify the patterns in transcription factor activity associated, and establish that the transcription factors TEAD and AP1 are specifically activated. Subsequently, they explore the mechanisms by which this occurs, which are different for each transcription factor. Finally, the authors identify that nuclear flattening mediates the known effects of ROC-mediated contractility, or cell spreading. The results of this manuscript reveal an important modulation of cell cycle progression by a purely physical parameter, nuclear shape, which is of great interest and may have potential major implications in several physiological scenarios. The work is carried out in an elegant, systematic, and rigorous way, and I therefore strongly support publication. I only have a set of minor comments to be addressed before publication.

Comment 1: My main comment is that even though the authors explore different mechanisms for the activation of YAP and c-Jun according to previously reported possibilities, it would be interesting to check the respective perturbations with the other candidate as well. That is, does lamin A/C depletion also affect YAP levels, and does importazole also somehow affect c-JUN phosphorylation? This would help to discriminate to what extent the mechanisms are indeed independent.

Our response: We thank the reviewer for his/her comment. We performed additional experiments and observed that lamin A/C depletion did not affect YAP nuclear entry, whereas treatment with importazole decreased nuclear c-Jun phosphorylation in response to nuclear flattening. These results indicate that 2 partly independent pathways act upstream TEAD and AP1 activation. We have added these results to the revised manuscript (sup figure 5f-g; sup 5b).

Comment 2: I find it interesting that LATS mutations affect YAP ratios in control cells, but does not impact the effect of forced nuclear flattening. A potential explanation for this would be that LATS mutations somehow affect nuclear shape per se, and this impacts YAP nuclear ratios aside from a direct effect of the hippo pathway. This would explain why if this nuclear deformation is forced exogenously, LATS has no effect. The authors could measure nuclear shapes in LATS-mutated cells to assess this, this would in fact be very interesting since it would help reconcile conflicting reports on the role of the hippo pathway in YAP mechanoresponses (an aspect which the authors themselves mention in the text).

Our response: As the reviewer, we too were surprised by this result. In the revised version of the manuscript we have included an analysis of nuclear morphology in cells transfected with LATS mutants and we observed no significant differences when compared to control cells (sup figure 5e).

Comment 3: In their discussion, the authors mention that "the impact of nuclear shape on gene expression was not known". This is an overstatement, as there are several works that analyse this from different perspectives (see for instance the work by the Shivashankar group in Singapore). This does not remove novelty from this very interesting work, but the authors should rephrase this to properly acknowledge previous work.

Our response: We replaced the sentence (and removed "was not known") to discuss previous studies reporting that changes in nuclear morphology can regulate transcription (including work

form Niethammer and colleagues, Shivashankar's group and Roca-Cusachs group).

Comment 4: As a minor aspect, the authors mention that in the case of YAP, inhibition of nuclear import decreased YAP translocation, but that a significant effect of nuclear flattening remained. This is then used to argue that this may not fully explain the results. Whereas it is true that a significant effect remains, it is extremely small, and actually the YAP ratios with blocked import are all almost exactly in the range of blebbistatin-treated cells. In this case, I would thus say that the differences in conditions of import inhibition are significant, but not biologically relevant. Thus, the authors' results are fully consistent with a regulation by nuclear import as previously reported. Again, I don't think this reduces the novelty of the work in any way - the current work goes well beyond previous findings in many ways - but this should be clarified.

Our response: We changed the text as suggested by the reviewer (and added "but not biologically relevant").

Minor Comment: the authors state that the agarose pads employed do not lead to nuclear herniation, but that data is not shown. Please show.

Our response: This result has been added to the revised version of the manuscript and shows no difference between control cells and cells compressed by the agarose pad (sup figure 1d).

2nd Editorial Decision

11 June 2019

Thank you for submitting the revised version of your manuscript. It has now been seen by all of the original referees.

As you can see, all referees find that the study is significantly improved during revision and recommend publication. Before I can accept the manuscript, I need you to address the below minor/editorial points:

- Referee #3 has some remaining concerns. In particular, he/she finds that to conclusively demonstrate the requirement of nuclear shape deformation for regulation of YAP and AP-1 activity at the G1/S transition, it should also be checked in the LINC loss of function conditions (SUN1/2 kd). He/she also recommends repeating some knockdown experiments that were performed with a single siRNA with independent siRNAs. I have discussed these points with referees #1 and 2 and we all came to the conclusion that these additional experiments are not required for publication. Earlier work from Roca Cusachs has already demonstrated that mechanical nuclear deformation induced nuclear YAP translocation does not require LINC. Therefore we do not think that it is necessary to test its requirement in this context as well.
-

Thank you again for giving us to consider your manuscript for EMBO Reports, I look forward to your revision.

REFEREE REPORTS

Referee #1:

In this revised version of their manuscript, Aureille and colleagues have addressed all my concerns. They have revised their writing to avoid overinterpretation of their results and better discussed prior knowledge. I think that their manuscript now describes accurately what their experiments show and I am still convinced of the broad interest of their findings. I thus recommend publication of the article without further revision.

Referee #2:

The authors have adequately addressed my concerns and in my view the manuscript is now ready for publication.

Referee #3:

The authors have extended the study of nuclear height during cell cycle progression to two more cell lines, as requested in the comments on the first version of the MS.

On the other hand, for most experiments involving RNAi the authors still display results obtained with a single transfection. Only for lamina A/C they report results obtained with two independent shRNAs (Suppl. Fig 5g). In the case of SUN, YAP and JUN it looks like they used a mix of two siRNAs transfected together in the same well/dish. For sure the authors will acknowledge that this is not equivalent to the independent transfection (= in different wells) of 2 or more siRNAs (or combinations) targeting the same mRNA, and showing that they independently produce the same results (which is essential in order to conclude that the observed results really depend on the absence of a protein, and they're not off-target effects of siRNA transfection).

Even more important, the definitive experiment proving that nuclear mechanotransduction (and not "global" cytoskeletal mechanotransduction) is specifically involved in the regulation of YAP and AP-1 activity at the G1/S transition is still missing: compression of blebbistatin treated cells with agarose pads should rescue YAP and JUN activation, YAP+JUN transcriptional activity and S-phase entry also in the absence of a functional LINC complex. For example, they should perform the experiment depicted in figure 1f with cells depleted of SUN1/2 vs control cells. This experiment will tell if nuclear shape can control S-phase entry independently of the connection with the cytoplasmic cytoskeleton. We believe that a similar result would be essential in a paper whose title is "Nuclear envelope deformation controls cell cycle progression in response to mechanical force", and much more relevant than any warning that it is impossible to exclude a role for cytoplasmic mechanotransduction.

So far, the authors have shown that Lamin A/C is required to rescue Jun phosphorylation in blebbi-treated compressed cells (whereas Lamin A/C depletion seems to reduce YAP nuclear localization in control cells, but it does not impair the effect of compression...). I'm afraid these additional data are not the answer to the question: "does nuclear flattening control cell cycle progression?". For the sake of clarity, I'm not asking for more mechanistic details, but a more definitive demonstration of what the title claims is mandatory for this article to be suitable for publication. Addressing this point would strengthen the key message of the MS.

2nd Revision - authors' response

18 June 2019

The authors performed all minor editorial changes.

3rd Editorial Decision

3 July 2019

Thank you for submitting your revised manuscript. I have now looked at everything and all is fine. Therefore I am very pleased to accept your manuscript for publication in EMBO Reports.

Corresponding Author Name: Christophe Guilluy

EMBO reports

EMBOR-2019-48084